perinatal mental health; low- and middle-income countries (LMICs); scale; sustainability; scoping review

**Corresponding author:**
Katie Rose M. Sanfilippo;
Email: katie-rose.sanfilippo@citystgeorges.ac.uk

# Strategies for spreading, scaling and sustaining perinatal mental health interventions in low- and middle-income countries (LMICs): A scoping review and thematic synthesis

Katie Rose M. Sanfilippo[1] , Musa Krubally[1,2], Melina Michelen[1], Gaotswake Patience Kovane[3], Lottie Anstee[4], Ifeyinwa Yusuf[5], Shanon McNab[6] and Simone Honikman[7]

[1]Department of Population Health & Policy, Centre for Health and Care Innovation Research (CHIR), City St George's University of London, London, UK; [2]Barts Health NHS Trust, London, UK; [3]NuMIQ Research Focus Area, Faculty of Health Sciences, North-West University, Mahikeng, South Africa; [4]School of Psychology, University of Roehampton, London, UK; [5]Consultant, Nigeria; [6]Consultant, Thailand and [7]Perinatal Mental Health Project, Department of Psychiatry and Mental Health, Centre for Public Mental Health, University of Cape Town, Cape Town, South Africa

## Abstract

Common perinatal mental health conditions are especially prevalent in low- and middle-income countries (LMICs) and are associated with numerous adverse effects. While complex interventions have been developed and tested, there has been limited exploration of how these interventions can be implemented and sustained at scale. This scoping review aims to explore the strategies discussed for scaling, spreading and sustaining complex perinatal mental health interventions in LMICs. We conducted a systematic search in APA PsycINFO, Cinahl, Medline (EBSCOhost), Embase, MIDIRS (Ovid Online) and ProQuest for reports published between January 2010 and November 2023, using search terms related to scaling innovations, perinatal mental health and LMICs. We also conducted a grey literature search using the websites of organisations that focus on maternal mental health. We identified 42 information sources. Using thematic synthesis, scale, spread and sustainability strategies regarding workforce diversity, integration of health services, tool and method development, adaptation, training, supervision and support and stakeholder engagement were identified. The study identified persistent gaps in the literature around how interventions move beyond early adaptation and implementation phases. These included the need for more consistency and shared understanding around terminology and increased interdisciplinary collaboration, especially drawing on fields such as implementation science. The findings from this review open new avenues for research and policy on expanding perinatal mental health interventions in LMICs, with an emphasis on long-term sustainability and interdisciplinary perspectives.

## Impact statement

There is a significant global burden of common perinatal mental health conditions, the most common complications of childbearing, with about one in five women being affected in LMICs. Health systems in these settings have not scaled, spread or sustained perinatal mental health services despite the evidence for the effectiveness of several interventions. This is, in part, due to a range of systems-level factors. Previous work on perinatal mental health interventions has been limited in scope regarding strategies to scale, spread and sustain perinatal mental health interventions in LMICs. Therefore, this scoping review provides a novel contribution by systematically examining the scale, spread and sustainability strategies of perinatal mental health interventions in LMICs from 42 information sources. The strategies identified are interconnected and used concurrently across the sources. These strategies included workforce diversity, integration into other health services, tool and method development, adaptation of existing interventions, training, supervision and support and stakeholder engagement. Unlike prior reviews, it includes both peer-reviewed and grey literature, allowing for a more comprehensive understanding of the evidence base. The review also identifies persistent gaps in the literature, particularly around how interventions move beyond early adaptation and implementation phases. Efforts to scale, spread and sustain perinatal mental health interventions in LMICs rely on these coordinated strategies to embed interventions within complex health systems and achieve equitable, long-term impact. Interdisciplinary perspectives and coordination across various stakeholders are essential. By organising evidence around concrete strategy types and highlighting where evidence remains thin, the review supports informed decision-making for programme design, funding and policy and provides a foundation for comparative, interdisciplinary implementation research to strengthen long-term sustainability at scale.

## Introduction

Common perinatal mental health conditions (CPMHCs), which occur during pregnancy and the year after birth, are the most common complications of childbearing worldwide (Howard and Khalifeh, 2020). Symptoms of CPMHCs, especially depression (~26%) and anxiety (~22–37%), present a significant global burden (Al-Abri et al., 2023; Aziz et al., 2025). Prevalence varies by measurement and perinatal period but remains high worldwide. Levels are consistently greater in low- and middle-income countries (LMICs), with perinatal depression around 24.7% and antenatal anxiety up to 29.2%, compared with lower estimates in high-income settings (Nielsen-Scott et al., 2022; Mitchell et al., 2023).

When untreated, these conditions substantially reduce quality of life and are associated with adverse maternal and neonatal outcomes, including preeclampsia, preterm birth, low birth weight and increased neonatal hospitalisation (Dadi et al., 2022; Li et al., 2022; Al-Abri et al., 2023). Disrupted mother–infant attachment is associated with longer-term implications for children's emotional regulation and behavioural functioning (Donald et al., 2018; Al-Abri et al., 2023). Maternal postnatal depression is associated with double the risk of anxiety and double the risk of depression in adolescent and young adult offspring (Chithiramohan and Eslick, 2023). Untreated CPMHCs generate wider economic costs through impaired functioning and reduced productivity (Bauer et al., 2022). Together, these pose significant family-level and intergenerational impacts. The effects are often amplified in LMICs, where social and structural determinants of mental health compound risk and limit access to care (Fisher et al., 2012; Kirkbride et al., 2024).

There exists a growing evidence base on how best to support women in these settings. Interventions that are integrated into routine maternal care have shown effectiveness (Prom et al., 2022). In fact, the World Health Organization (WHO) advocates for integrating psychological support into routine maternal care through, for example, the Mental Health Gap Action Programme (mhGAP) (WHO, 2016), which provides non-specialists with the clinical protocols needed to treat mental health conditions in primary care settings. Research has shown that task-sharing of psychosocial interventions delivered by non-specialists is effective in prevention as well as reduction of symptoms (Prina et al., 2023). These interventions are delivered in a range of healthcare settings, homes and community centres, and include techniques derived from cognitive behavioural therapy (CBT), problem-solving therapy (PST), interpersonal therapy (IPT) and psychoeducation (Prina et al., 2023).

One example of a task-sharing intervention is the Thinking Healthy Programme (THP). The THP is a WHO-endorsed, low-intensity psychological intervention that uses CBT techniques, which are simplified for delivery by non-specialist community health workers to treat perinatal depression. It has become a global priority within the WHO's mhGAP framework because it offers an evidence-based, culturally adaptable model that can be integrated into routine maternal care. Shortfalls in comprehensive intervention design literature in LMICs include a lack of evidence for increased treatment intensity for more severe illness, pharmacotherapy and non-mental health professional training and supervision (Prom et al., 2022).

There is a large treatment gap for mental healthcare in general, where more than 90% of individuals with mental, neurological or substance use disorders are not receiving adequate care (Demyttenaere et al., 2004; Wagenaar et al., 2022). This may be due to the strain on health systems in LMICs to provide a balance of care for the high rates of communicable diseases and non-communicable diseases,

such as depression (Collins et al., 2013), as well as a general scarcity of trained mental health workers (Saxena et al., 2007). Further factors include reduced foreign aid and multi-level stigma that constrains policy development and service delivery (Patel et al., 2018). In perinatal mental health, the intersection of gender with HIV, gender-based violence and poverty further marginalises affected women, limiting political prioritisation, sustained financing and integration of services for high-risk populations (Wilson et al., 2024). Where services do exist, these have not been sustained, scaled or effectively spread (Nillni and Gutner, 2019; McNab et al., 2022). This may be due to limited available learning from programmes being implemented in real-world settings and the problematic translation of interventions into scalable and sustainable programmes in LMICs (Nillni and Gutner, 2019; McNab et al., 2022). This problematic translation may be due to many interacting factors such as limited budgets and a lack of dedicated mental health funding, a shortage of trained health workers to provide care, the exclusion of mental health from broader global maternal and child health initiatives and the compounding impact of poverty and gender-based violence on a woman's ability to seek help (McNab et al., 2022).

Following Côté-Boileau et al.'s (2019) definitions, for this review we define scale as 'the ambition or process of expanding the coverage of health interventions'. Spread is defined as 'the process through which new working methods developed in one setting are adopted, perhaps with appropriate modifications, in other organisational contexts' (Côté-Boileau et al., 2019). Sustainability is defined as 'the process through which new working methods, performance enhancements and continuous improvements are maintained for a period appropriate to a given context' (Côté-Boileau et al., 2019).

Previous work on perinatal mental health interventions has been limited in scope regarding strategies to scale, spread and sustain these interventions in LMICs. Relevant research includes a recent book chapter providing a broad overview of strategies used to scale perinatal mental health interventions (Waqas and Rahman, 2023) and a recent landscape analysis and implementation guide that described how example interventions were delivered, but not how they expanded or spread across contexts (McNab et al., 2022; WHO, 2022). A 2011 review summarised strategies used for the spread and scale of complex mental health interventions in LMICs, and a recent WHO guidance document lays out strategies for scaling in public health systems generally, but neither includes a focus on perinatal mental health (Eaton et al., 2011; WHO, 2026).

This scoping review aims to equip researchers, policymakers and implementers with a synthesis of strategies that have been used to scale, spread and sustain perinatal mental health interventions in LMICs. This work is addressing a critical gap in the literature, which has largely focused on intervention effectiveness rather than implementation trajectories. The review consolidates strategies across peer-reviewed and grey sources to identify actionable approaches and persistent bottlenecks. By organising evidence around concrete strategy types and highlighting where evidence remains thin, the review supports informed decision-making for programme design, funding and policy and provides a foundation for comparative, interdisciplinary implementation research to strengthen long-term sustainability at scale.

## Methods

### Search strategy and selection criteria

We conducted a scoping review. This methodology is particularly useful in disciplines with evolving evidence as it examines broader

areas, delves beyond the effectiveness and experience of an intervention and can map literature to locations, sources, methods and origin (Peters et al., 2015). This scoping review was guided by the Preferred Reporting Items for Systematic Reviews and Meta-Analyses extension for scoping reviews (PRISMA-ScR) checklist (Tricco et al., 2018), as outlined in Supplementary Material 1, and the scoping review framework outlined in the Joanna Briggs Institute (JBI) Reviewer's Manual (Peters et al., 2020).

Table 1 details the eligibility criteria for the included information sources. We searched the following electronic databases on November 7, 2023: APA PsycINFO, Cinahl, Medline (EBSCOhost), Embase, MIDIRS (Ovid Online) and ProQuest. Search terms were developed based on terminology for scaling innovations (Côté-Boileau et al., 2019), perinatal mental health and LMICs. Data were limited from 2010 onwards to follow up on the review by Eaton et al. (2011). The search strategy for Medline is available in Supplementary Material 3 and was adapted for the other databases by the research team with guidance from a librarian.

Further, we conducted a grey literature search using the websites of organisations that focus on perinatal mental health (Supplementary Material 2). For each website, we manually searched publication libraries, resource pages and reports using the same core terms. Grey literature included organisational reports, policy guidance, programme descriptions and other non-peer-reviewed materials available through these sites.

**Table 1.** Eligibility criteria for the information sources

|  | Inclusion criteria | Exclusion criteria |
|---|---|---|
| Population | Women and girls who are pregnant or are up to a year after childbirth | Women outside of the perinatal period |
| Intervention | Strategies used for scaling, spreading and/or sustaining complex perinatal mental health interventions. This includes descriptions of strategies used for a discrete intervention or in a general context | Interventions related to adoption, preconception, abortion, miscarriage, stillbirth or ectopic pregnancies were not considered |
| Context | LMIC settings as defined by the World Bank classification (2025) | HICs |
| Outcome | Implementation, scaling or spreading outcomes. Information sources on adaptation methods were included where enough detail was described within the text. Information sources that described the barriers or enablers to a specific scaling strategy were also included | Outcomes focused on barriers to the implementation of a specific intervention within a specific context. Efficacy or cost-effectiveness outcomes of a specific intervention |
| Type of information source | No restrictions on study design. Non-empirical, empirical and grey literature with full texts available were included |  |
| Language | Published in English |  |

*Note*: LMIC, low- and middle-income countries.

## Data analysis

### Study selection

We used Rayyan (Ouzzani et al., 2016) for the study selection process. Two reviewers independently screened all records in two stages using the specified eligibility criteria, first by title and abstract and then by full text. Any disagreements were resolved through discussion or by involving a third independent reviewer.

### Data extraction

Two reviewers independently extracted data from the included sources using a data extraction sheet adapted from JBI, which was piloted with three sample sources before finalising it (Supplementary Material 4). Any discrepancies in data extraction were resolved through discussion or by involving a third independent reviewer. The data extracted included study characteristics (author, title, year, database/grey literature, study design, aim and context/country), intervention details (intervention type and target population) and strategies described (process, aim of the strategy, challenges and lessons learnt).

### Data appraisal

Following scoping review methodology, we did not undertake a quality appraisal or risk of bias assessment of the evidence in this review (Tricco et al., 2018; Peters et al., 2020).

### Synthesis of results

We first summarise the results descriptively, following guidance from Lockwood et al. (2019). The final results were reported through a thematic synthesis (Thomas and Harden, 2008) focused specifically on strategies for scale, spread and sustainability. This analysis included line-by-line coding of primary study findings, developing descriptive themes and then generating analytical themes.

## Results

The PRISMA flow diagram (Tricco et al., 2018) reports an overview of our search and selection process (Figure 1).

### Characteristics of the included information sources

Figure 2 describes the number of information sources by year and region. Websites accounted for three of the information sources identified (Mental Health Innovation Network, n.d.a, n.d.b; Perinatal Mental Health Project, n.d.) and, therefore, had no publication date.

Of the 42 information sources, 32 were identified through database searches and 10 through grey literature and citation review. Thirteen information sources did not discuss a specific intervention for perinatal mental health or reviewed multiple interventions. The Thinking Healthy Programme (THP) was the most frequently discussed intervention ($n = 19$). The remaining information sources discussed either integrating screening and support into routine care ($n = 5$), motivational interviewing and problem-solving therapy ($n = 2$) or other interventions ($n = 3$). See Table 2 for further details on the interventions discussed across the information sources.

### Scale, spread and sustainability strategies

A total of 25 information sources discussed more than one of the three strategies (spread, scale and sustainability) simultaneously. Thirty-four information sources discussed strategies to scale

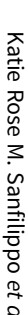

**Identification of studies via databases and registers**

**Identification of studies via other methods**

**Identification**

Records identified from:
Databases (n = 5,002)

Medline = 3500
CINHAL = 664
PsycInfo = 465
Embase = 661
MIDIRS = 31
Proquest= 41

Records removed *before screening*:
Duplicate records removed (n = 1261)

Records identified from:
Websites (n = 1)
Organisations (grey lit.) (n = 23)
Citation searching (n = 2)

**Screening**

Records screened
(n = 3,741)

Records excluded
(n = 3620)

Reports sought for retrieval
(n = 121)

Reports not retrieved
(n = 1 )

Reports sought for retrieval
(n = 25)

Reports not retrieved
(n = 1)

Reports assessed for eligibility
(n =120)

Reports excluded reasons:
Wrong outcome (n = 47)
Wrong setting (n = 8)
Wrong strategy/implementation
(n = 8)
Wrong publication (n =15)
Duplicate (n = 10)

Reports assessed for eligibility
(n = 24)

Reports excluded reason:
Not focused on spread/scale
methods (n = 9)
Not focused on
perinatal/maternal mental health
(n = 3)
Not focused on LMICs (n = 3)

**Included**

Studies included in review
(n = 42)

**Figure 1.** PRISMA flow chart. PRISMA, Preferred Reporting Items for Systematic Reviews and Meta-Analyses. Adapted from Page et al. (2021). For more information, visit: http://www.prisma-statement.org/.

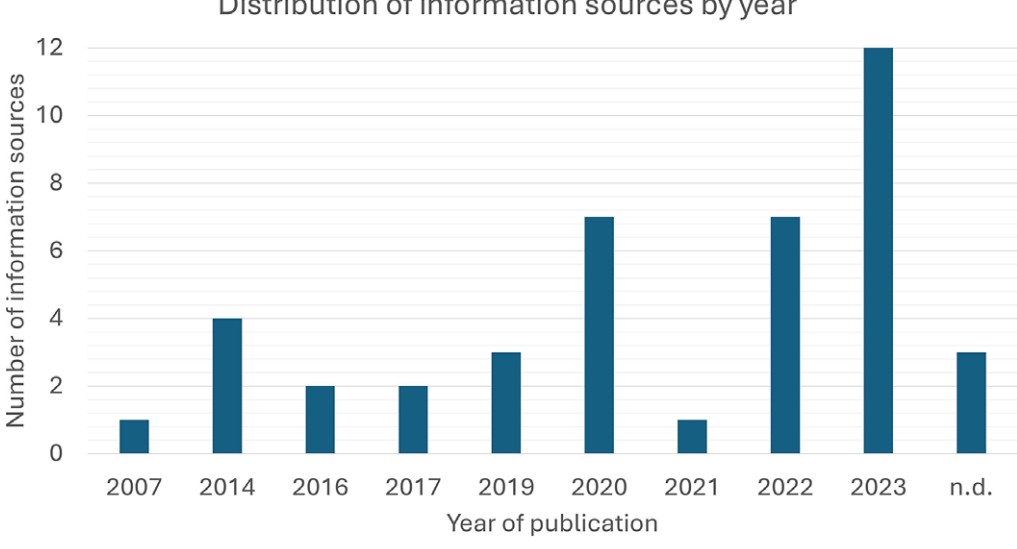

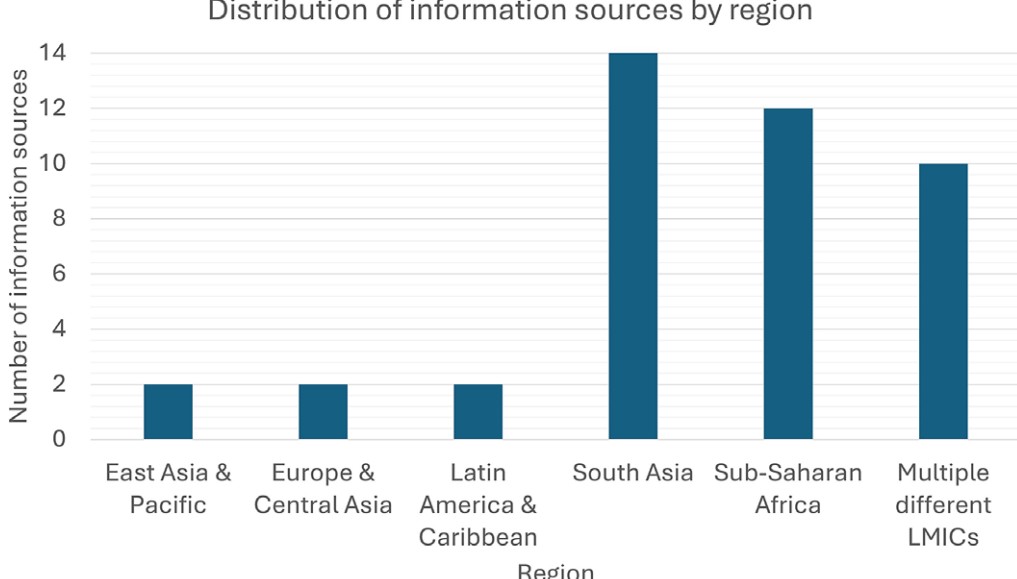

**Figure 2.** Information sources by year and region.

perinatal mental health interventions in LMICs, 17 information sources discussed spread strategies and 20 information sources discussed sustainability strategies. Definitions of what was meant by the terms 'scale', 'spread' and 'sustain' were not included in the information sources. None of the identified sources specifically used the term 'spread' when discussing how interventions were adapted and trialled in a new context. However, many information sources discussed adaptation methods, which we see as a primary spread mechanism within this body of literature.

### Thematic synthesis

Using thematic synthesis, we developed themes to describe the strategies used by the 42 information sources to scale, spread or sustain perinatal mental health interventions. We have categorised the themes into three strategic types: scale, spread and sustainability. This classification reflects how these strategies were primarily

discussed as mechanisms that support one of these overall processes. However, it is important to note that some strategies may contribute to more than one category. For example, stakeholder engagement is placed under sustainability, as it was mainly discussed in that context. Nevertheless, this strategy could also play a significant role in supporting both scale and spread. Additionally, even though we present the strategies as discrete themes, they are interconnected and interrelated. Many of the information sources described more than one strategy and used them simultaneously to support the scale, spread and/or sustainability of complex perinatal mental health interventions in LMICs.

### Scale strategies

#### Diversify workforce

Twenty information sources included the 'diversify workforce' scale strategy (Perinatal Mental Health Project, n.d.; Mental Health

**Table 2.** Included information sources

| References | Title | Source | Country | Intervention | Scale/spread/sustainability | Strategy/strategies discussed |
|---|---|---|---|---|---|---|
| Agampodi et al. (2023) | Incorporating early pregnancy mental health screening and management into routine maternal care: experience from the Rajarata Pregnancy Cohort (RaPCo), Sri Lanka | Database | Sri Lanka | Rajarata Pregnancy Cohort (antenatal screening approach and support package) | Scale | Diversify workforce/integration |
| Ahmad et al. (2020) | Measuring the implementation strength of a perinatal mental health intervention delivered by peer volunteers in rural Pakistan | Database | Pakistan | The Thinking Healthy Programme delivered by Peer Volunteers (THPP) | Scale | Tool and method development |
| Atif et al. (2016) | Barefoot therapists: barriers and facilitators to delivering maternal mental health care through peer volunteers in Pakistan: a qualitative study | Database | Pakistan | The THPP | Scale | Diversify workforce |
| Atif et al. (2017) | Mother-to-mother therapy in India and Pakistan: adaptation and feasibility evaluation of the peer-delivered Thinking Healthy Programme | Database | Pakistan & India | Adaptation of the Thinking Health Programme (THP) to be delivered by peers | Scale/Spread/Sustainability | Diversify workforce/Adaptation/Training, supervision and support |
| Atif et al. (2019a) | Delivering maternal mental health through peer volunteers: a 5-year report | Database | Pakistan | The THPP | Scale/Sustainability | Diversify workforce/Training, supervision and support |
| Atif et al. (2019b) | Scaling-up psychological interventions in resource-poor settings: training and supervising peer volunteers to deliver the 'Thinking Healthy Programme' for perinatal depression in rural Pakistan | Database | Pakistan | Training and supervision to sustain the THPP | Scale/Sustainability | Diversify workforce/Training, supervision and support |
| Atif et al. (2020) | Development of a psychosocial intervention to address anxiety during pregnancy in a low-income country | Citation review | Pakistan | Adaptation of the THP for perinatal anxiety | Spread | Adaptation |
| Atif et al. (2022) | Technology-assisted peer therapy: a new way of delivering evidence-based psychological interventions | Database | Pakistan | Technology-assisted version of the THPP | Scale/Sustainability | Diversify workforce/Tool and method development/Training, supervision and support |
| Atif et al. (2023) | Non-specialist-delivered psychosocial intervention for prenatal anxiety in a tertiary care setting in Pakistan: a qualitative process evaluation | Database | Pakistan | Adaptation of the THP for perinatal anxiety | Scale | Diversify workforce/Integration |
| Bakare et al. (2017) | Improving access to interventions among mothers screened positive for post-partum depression (PPD) at the National Programme on Immunization (NPI) clinics in south-western and south-eastern Nigeria – a service development report | Database | Nigeria | Service innovation integrating perinatal mental health services into primary healthcare | Scale | Integration |
| Bitew et al. (2022) | Adapting an intervention of brief problem-solving therapy to improve the health of women with antenatal depressive symptoms in primary healthcare in rural Ethiopia | Database | Ethiopia | MI-PST (motivational interviewing and problem-solving therapy) in primary care by peer counsellors | Spread | Adaptation |
| Boran et al. (2023a) | Delivering the Thinking Healthy Programme as a universal group intervention integrated into routine antenatal care: a randomized-controlled pilot study | Database | Turkey | Online brief group version of the THP, designed as a preventative intervention | Scale/Spread | Diversify workforce/Integration/Tool and method development/Adaptation |
| Boran et al. (2023b) | Adaptation and integration of the Thinking Healthy Programme into pregnancy schools in Istanbul, Turkey | Citation review | Turkey | Adaptation of the THP for group antenatal preventative care | Scale/Spread | Integration/Tool and method development/Adaptation |

*(Continued)*

**Table 2.** (*Continued*)

| References | Title | Source | Country | Intervention | Scale/spread/sustainability | Strategy/strategies discussed |
|---|---|---|---|---|---|---|
| Davies et al. (2019) | Psychotherapy for perinatal mental disorders in low-and middle-income countries | Database | LMICs | NA | Scale/Spread/Sustainability | Diversify workforce/Integration/Adaptation/Stakeholder engagement |
| Davies et al. (2022) | Implementation of a task-shared psychosocial intervention for perinatal depression in South Africa: a grounded theory process evaluation | Database | South Africa | AFFIRM-SA (a task-shared psychosocial counselling intervention) delivered by community health workers | Scale/Sustainability | Diversify workforce/Training, supervision and support/Stakeholder engagement |
| Fisher et al. (2014) | Translation, cultural adaptation and field testing of the Thinking Healthy Programme for Vietnam | Database | Vietnam | Adaptation of the THP within universal programmes for population health | Spread | Adaptation |
| Green et al. (2020) | Expanding access to perinatal depression treatment in Kenya through automated psychological support: development and usability study | Database | Kenya | Adaptation of the THP to an artificial intelligence system | Scale/Spread | Tool and method development/Adaptation |
| Honikman and Field (2020) | Maternal mental health in South Africa and the opportunity for integration | Database | South Africa | PMHP (an adaptable stepped-care model for mental health screening, counselling and case management) | Scale/Sustainability | Integration/Training, supervision and support/Stakeholder engagement |
| Jabeen et al. (2022) | Evidence of mobile health integration into primary health care systems for better maternal mental health in LMICs during the COVID-19 pandemic – review. | Database | LMICs | NA | Scale | Integration/Tool and method development |
| Keynejad et al. (2023) | Adapting brief problem-solving therapy for pregnant women experiencing depressive symptoms and intimate partner violence in rural Ethiopia | Database | Ethiopia | MI-PST delivered in primary care by peer counsellors | Spread | Adaptation |
| Kohrt et al. (2022) | The use of formative research to culturally adapt a psychosocial support programme for perinatal Mayan women in Guatemala | Database | Guatemala | Adaptation of the THP | Spread | Adaptation |
| Le et al. (2023) | Preventing perinatal depression: cultural adaptation of the Mothers and Babies Course in Kenya and Tanzania | Database | Kenya & Tanzania | Mothers and Babies Course (a cognitive behavioural intervention for women at risk of perinatal depression) | Scale/Spread | Integration/Adaptation |
| Manolova et al. (2023) | Integrating perinatal mental healthcare into maternal and perinatal services in low- and middle-income countries | Database | LMICs | NA | Scale/Sustainability | Diversify workforce/Integration/Tool and method development/Training, supervision and support/Stakeholder engagement |
| Marley et al. (2023) | Maternal mental health in Sub-Saharan Africa: a systematic review of interventions for common perinatal mental health disorders | Database | LMICs | NA | Scale/Spread | Diversify workforce/Adaptation |
| McNab et al. (2021) | A landscape analysis: the silent burden: common perinatal mental disorders in low- and middle-income countries | Grey literature | LMICs | NA | Scale/Sustainability | Diversify workforce/Integration/Tool and method development/Training, supervision and support/Stakeholder engagement |
| Mental Health Innovation Network (n.d.a) | Scale-up of a maternal depression intervention through technology in a post-conflict area | Grey literature | Pakistan | The THP in areas affected by humanitarian crises | Scale/Sustainability | Tool and method development/Training, supervision and support |
| Mental Health Innovation Network (n.d.b) | Thinking Healthy pilot in Peru | Grey literature | Peru | Adaptation of the THP | Scale/Spread/Sustainability | Diversify workforce/Adaptation/Training, supervision and support/Stakeholder engagement |

(*Continued*)

**Table 2.** (*Continued*)

| References | Title | Source | Country | Intervention | Scale/ spread/ sustainability | Strategy/strategies discussed |
|---|---|---|---|---|---|---|
| Nisar et al. (2020) | Making therapies culturally relevant: translation, cultural adaptation and field-testing of the Thinking Healthy Programme for perinatal depression in China | Database | China | Adaptation of the THP | Spread | Adaptation |
| Obonyo et al. (2023) | Diverse policymaker perspectives on the mental health of pregnant and parenting adolescent girls in Kenya: considerations for comprehensive, adolescent-centred policies and programmes | Database | Kenya | NA | Sustainability | Stakeholder engagement |
| Owais et al. (2023) | Integrating maternal depression care at primary private clinics in low-income settings in Pakistan: a secondary analysis | Database | Pakistan | Maternal depression component of an early child development (ECD) intervention | Scale/ Sustainability | Integration/Stakeholder engagement |
| The Partnership for Maternal, Newborn, and Child Health (2014) | Maternal mental health: why it matters and what countries with limited resources can do | Grey literature | LMICs | NA | Scale/ Sustainability | Integration/Stakeholder engagement |
| Perinatal Mental Health Project (n.d.) | Perinatal Mental Health Project: to address the treatment gap for perinatal mental disorders in South Africa | Grey literature | South Africa | PMPH (an adaptable stepped-care model for mental health screening, counselling and case management) | Scale/ Sustainability | Diversify workforce/ Integration/Training, supervision and support/ Stakeholder engagement |
| PRIME (2016) | PRIME (Programme for Improving Mental Health Care): evidence on scaling-up mental health services for development | Grey literature | Ethiopia, India, Nepal, South Africa & Uganda | NA | Scale/ Sustainability | Integration/Stakeholder engagement |
| Sarkar et al. (2022) | Integration of perinatal mental health care into district health services in Uganda: why is it not happening? The Four Domain Integrated Health (4DIH) explanatory framework | Database | Uganda | NA | Scale/ Sustainability | Integration/Tool and method development/Stakeholder engagement |
| Singla et al. (2014) | 'Someone like us': delivering maternal mental health through peers in two South Asian contexts | Database | Pakistan & India | The THPP | Scale | Diversify workforce |
| Suchman et al. (2020) | Mothering from the inside out: adapting an evidence-based intervention for high-risk mothers in the Western Cape of South Africa | Database | South Africa | Mothering from the Inside Out (a mentalisation intervention) | Spread | Adaptation |
| UNFPA/WHO (2009) | Maternal mental health and child health and development in resource-constrained settings | Grey literature | LMICs | NA | Scale/ Sustainability | Integration/Stakeholder engagement |
| Waqas and Rahman (2023) | Innovations in scaling up interventions in low- and middle-income countries: parent-focused interventions in the perinatal period and promotion of child development | Database | LMICs | NA | Scale/ Sustainability | Diversify workforce/ Integration/Tool and method development/Training, supervision and support |
| Waqas et al. (2022) | Predicting remission among perinatal women with depression in rural Pakistan: a prognostic model for task-shared interventions in primary care settings | Database | Pakistan | Developing the THP using a tool to predict the prognosis of perinatal depression | Scale | Tool and method development |
| WHO (2022) | Guide for integration of perinatal mental health in maternal and child health services | Grey literature | LMICs | NA | Scale/ Spread/ Sustainability | Diversify workforce/ Integration/Adaptation/ Training, supervision and support |

**Table 2.** (*Continued*)

| References | Title | Source | Country | Intervention | Scale/spread/sustainability | Strategy/strategies discussed |
|---|---|---|---|---|---|---|
| Zafar et al. (2014) | Integrating maternal psychosocial well-being into a child-development intervention: the five-pillars approach | Database | Pakistan | Adaptation of the THP integrated into a nutrition and early child development programme | Scale/Spread | Diversify workforce/Integration/Adaptation |
| Zhu et al. (2022) | Factors affecting the implementation of task-sharing interventions for perinatal depression in low- and middle-income countries: a systematic review and qualitative metasynthesis | Database | LMICs | NA | Scale | Diversify workforce |

*Note*: AFFIRM-SA, task-shared psychosocial counselling intervention; LMICs, low- and middle-income countries; MI-PST, motivational interviewing and problem-solving therapy; NA, no specific intervention or multiple interventions included; PMHP, Perinatal Mental Health Project; THP, Thinking Healthy Programme; THPP, Thinking Healthy Programme – Peer Delivery; UNFPA, United Nations Population Fund; WHO, World Health Organization.

Innovation Network, n.d.b; Singla et al., 2014; Zafar et al., 2014; Atif et al., 2016; Atif et al., 2017; Davies et al., 2019; Atif et al., 2019a, 2019b; McNab et al., 2021; Atif et al., 2022; Davies et al., 2022; WHO, 2022; Zhu et al., 2022; Agampodi et al., 2023; Atif et al., 2023; Manolova et al., 2023; Marley et al., 2023; Waqas and Rahman, 2023; Boran et al., 2023a). This theme encompasses strategies to increase the capacity of perinatal mental health services through training those who are not mental health specialists to deliver interventions. This strategy was commonly described as 'task-sharing' or 'task-shifting'. Task-sharing is defined as 'extending the types of health providers who can deliver health services appropriately' (WHO, 2022). Task-shifting is when 'tasks are redistributed from highly qualified providers to health providers with less intensive training to make more efficient use of human resources' (WHO, 2022). Task-sharing and task-shifting strategies typically operate together to support the effective scaling of interventions in contexts with health workforce shortages and stigma related to mental health service use (Davies et al., 2019). The delivery of complex perinatal mental health interventions was often shifted to peer volunteers or lay health workers with fewer qualifications and greater connections with local communities, while task-sharing enabled healthcare professionals to offer training, supervision and support for more specialist cases (e.g., Atif et al., 2016, 2019a, 2019b).

Four main groups of non-specialists were discussed (non-specialist providers such as primary healthcare staff, community health workers [CHWs], lay counsellors and peers). One example of training non-specialist providers included Agampodi et al. (2023). They described a two-stage screening approach for antenatal anxiety and depression developed in Sri Lanka, where initial screening was carried out by trained primary healthcare workers. By upskilling non-specialist providers to conduct simple perinatal mental health screening, they found that services could detect more women in need of support and reduce the burden on specialist mental health services. Davies et al. (2022) conducted an evaluation to explore the processes that occurred during the delivery of a task-shared psychosocial counselling intervention for perinatal depression by CHWs in South Africa. The authors recognised the success of task-sharing strategies in various populations but highlighted the importance of considering factors that may compromise the quality and efficacy of the intervention. These included the influence of context (e.g., socio-economic factors undermining the improvement in mood through their consistent presence in the women's lives) and counselling factors (e.g.,

CHWs' misinterpretation of therapeutic modalities and inappropriate reading from the manual).

Training peer women to deliver mental health services was discussed in interventions across different LMICs. Atif et al. (2016) described the justification for training peers to deliver THP in Pakistan and how it helped address significant workload barriers for the Lady Health Workers (LHWs), similar to CHWs in other contexts. Often LHWs have very high workloads and are responsible for all communicable diseases, so are unable to incorporate mental health into their routine work. The workload and competing priorities of LHWs can necessitate the use of peers to provide perinatal mental health services.

*Integration*

Twenty information sources included the scaling strategy 'integration' (Perinatal Mental Health Project, n.d.; UNFPA/WHO, 2009; The Partnership for Maternal, Newborn, and Child Health, 2014; Zafar et al., 2014; PRIME, 2016; Bakare et al., 2017; Davies et al., 2019; Honikman and Field, 2020; McNab et al., 2021; Atif et al., 2022; Jabeen et al., 2022; Sarkar et al., 2022; WHO, 2022; Agampodi et al., 2023; Atif et al., 2023; Manolova et al., 2023; Owais et al., 2023; Waqas and Rahman, 2023; Boran et al., 2023a, 2023b). This theme refers to the integration of interventions into other care pathways such as primary care, maternal and child health (MCH) or community models of care. One example of this is a stepped-care approach where low-intensity and less resource-intensive interventions are offered to most women and high-intensity interventions are saved for those with more severe mental health needs. Following the principles of the stepped-care approach, Agampodi et al. (2023) described the integration of perinatal mental health screening in primary care services in Sri Lanka, where screening was carried out by trained primary healthcare workers as part of the existing pathway of perinatal care. Where appropriate, women where then referred to public health midwives or a psychiatrist, depending on the level of risk.

Similarly, the Perinatal Mental Health Project (https://pmhp.za.org/, n.d.) in South Africa is a comprehensive stepped-care mental health service for pregnant women integrated into MCH services. During the first antenatal visit, women are given a three-item mental health screening questionnaire by clinic staff or counsellors, and women who screen positive for depression, anxiety and/or suicidality are referred to a counsellor. The counsellor conducts a brief 'engage, assess, triage' session with them to assess symptom severity, risk factors and to assign women to appropriate

levels of care. The stepped-care approach was discussed across many information sources as a helpful and successful way to increase maternal mental health care and support.

Integrating into a community setting is discussed by Boran et al. (2023b). They describe how the THP was adapted and integrated into pregnancy schools (online classes) in Istanbul, Turkey. By integrating perinatal mental health services in community settings, the authors aimed to improve the accessibility and parity of care, especially for disadvantaged populations who may not be adequately supported by existing healthcare services.

### Tool and method development

Twelve information sources included what we describe as the 'tool and method development' scale strategy (Mental Health Innovation Network, n.d.a, n.d.b; Ahmad et al., 2020; Green et al., 2020; McNab et al., 2021; Atif et al., 2022; Jabeen et al., 2022; Sarkar et al., 2022; Waqas et al., 2022; Manolova et al., 2023; Waqas and Rahman, 2023; Boran et al., 2023a). This strategy focuses on the development of new tools or methods to support the scale of interventions. These include technological or digital solutions and implementation tools or method development. One example of technological tool development is a study from Kenya, where an artificial intelligence (AI) tool was developed to deliver an adapted version of the THP (Healthy Moms) through a chatbot (Green et al., 2020). The authors reported that most of the women who tried the chatbot demonstrated a positive attitude towards the service and expressed trust in the chatbot, with some participants finding the privacy of chatting with a machine better than speaking to a counsellor. However, some challenges included ensuring that the content was engaging and the service was able to cope with misunderstandings.

Information sources, especially those written more recently, discussed the importance of drawing on theories and frameworks from implementation science (McNab et al., 2021; Davies et al., 2022). For example, McNab et al. (2021) used the Consolidated Framework for Implementation Research (CFIR) to identify how different components, such as policy environments and cultural norms, dictate the success of task-sharing interventions. Meanwhile, Davies et al. (2022) argue that for perinatal mental health interventions to transition from pilot projects to national programmes, research must move beyond clinical efficacy to prioritise implementation outcomes like acceptability and sustainability. Ahmad et al. (2020) developed an implementation tool to be used to gauge the level of implementation strength of the Thinking Healthy Programme Peer-delivered (THPP) in Pakistan. The key components of the tool included the competence of peer volunteers, the supervision attended and the number and duration of THPP sessions. The authors suggested that these types of tools could be used by policymakers and local governments to monitor implementation strength and impact on clinical outcomes.

In Uganda, Sarkar et al. (2022) developed a Four Domain Integrated Health (4DIH) explanatory framework, including: (1) Nature of the health problem; (2) State of the formal health system and its various components; (3) Additional, alternative and pluralistic systems of care and support; and (4) Global priorities, programmatic concerns and resource allocation. They argue that combining these domains with participatory methods and stakeholder engagement may help to identify current shortcomings and possible avenues for change, as well as potential local and global considerations. By mapping these interconnected domains, the framework acts as a strategy to facilitate scaling by ensuring that interventions are not only clinically sound but also structurally compatible with both local informal networks and international funding cycles, preventing 'pilotitis' (Scarbrough et al., 2024) that often stalls health initiatives.

### Spread strategies

#### Adaptation

Seventeen information sources included the 'adaptation' spread strategy (Mental Health Innovation Network, n.d.b; Fisher et al., 2014; Zafar et al., 2014; Atif et al., 2017; Davies et al., 2019; Atif et al., 2020; Green et al., 2020; Nisar et al., 2020; Suchman et al., 2020; Bitew et al., 2022; Kohrt et al., 2022; WHO, 2022; Keynejad et al., 2023; Le et al., 2023; Marley et al., 2023; Boran et al., 2023a, 2023b). This theme includes strategies used to adapt an intervention to a new context, population or condition. Most information sources use the strategy of adaptation when spreading to a new context. Keynejad et al. (2023) described how a brief problem-solving intervention was adapted for pregnant women experiencing depressive symptoms and intimate partner violence (IPV) in rural Ethiopia. Kohrt et al. (2022) discussed in detail how they adapted the THP for use within a community health organisation serving indigenous Tz'utujil Mayan families in Guatemala.

Some information sources discussed adapting the intervention to serve a different population or condition. For example, the THP was originally developed to support women experiencing perinatal depression, but Boran et al. (2023b) adapted the THP to a group-based universal and preventative intervention delivered to all women as part of their routine antenatal care. In an RCT testing the efficacy of the adapted intervention, the researchers found that most women felt the intervention was beneficial as it provided them with the opportunity to share and learn from each other's experiences (Boran et al., 2023a).

Various adaptation methods were described and frameworks used, as detailed in Table 3. Patient and public involvement was described as an important component of any adaptation process. Suchman et al. (2020) adapted a parenting intervention developed in the United States for use in South Africa, highlighting how adaptation can support spread across contexts. The intervention, Mothering from the Inside Out (MIO), is a short-term, 12-session adjunctive therapy for vulnerable mothers provided in tandem with other medical and/or mental health services. It uses 'mentalising' to help mothers regulate their own emotions, which in turn creates a foundation for their children to develop healthy emotional control and secure attachments. The authors reflected on the importance of building relationships with community members as equal partners during intervention adaptation, centralising their understanding of the local culture and needs of the target population.

### Sustainability strategies

#### Training, supervision and support

Thirteen information sources included the 'training, supervision and support' sustainability strategy (Perinatal Mental Health Project, n.d.; Mental Health Innovation Network, n.d.a, n.d.b; Atif et al., 2017, 2019a, 2019b; 2022; Honikman and Field, 2020; McNab et al., 2021; Davies et al., 2022; WHO, 2022; Manolova et al., 2023; Waqas and Rahman, 2023). This theme includes strategies for the training and support of those who are facilitating interventions, especially those who are not mental health specialists, with a focus on how this can sustain the interventions long term. Training and supervision were discussed as helping to avoid voltage drop (the intervention loses some degree of its potency or fidelity when

**Table 3.** A list of the key adaptation frameworks and methods used in each information source

| References | Adaptation frameworks | Adaptation methods |
|---|---|---|
| Atif et al. (2017) | Theory of Change | Focus groups, interviews, workshops |
| Atif et al. (2020) | Medical Research Council (MRC)/National Institute for Health and Care Research (NIHR) complex intervention framework | Focus groups, interviews, evidence review |
| Bitew et al. (2022) | MRC/NIHR complex intervention framework, ADAPT, ecological validity model (ECM), Theory of Change | Participatory methods |
| Boran et al. (2023a) | Bernal Framework for adaptation | Stakeholder engagement, field testing, focus groups, interviews |
| Boran et al. (2023b) | Bernal Framework for adaptation | Stakeholder engagement, field testing, focus groups, interviews |
| Davies et al. (2019) | | Stakeholder engagement |
| Fisher et al. (2014) | WHO's Integrated Management of Childhood Illness (IMCI) Adaptation Guide | Translation, field testing, stakeholder engagement |
| Green et al. (2020) | | Field testing, interviews, questionnaires |
| Keynejad et al. (2023) | ADAPT, NIHR context guidance, MRC/NIHR complex intervention framework, Theory of Change | Interviews, desk review |
| Kohrt et al. (2022) | Cultural Adaptation of Scalable Psychological Interventions framework | Interviews, participatory methods, questionnaires, stakeholder engagement |
| Le et al. (2023) | Updated Framework for Reporting Adaptations and Modifications-Enhanced (FRAME) | Interviews, field observations, field visits, reflection meetings |
| Marley et al. (2023) | | Interviews, focus groups, stakeholder engagement |
| Mental Health Innovation Network (n.d.b) | | Training workshops, field testing, stakeholder engagement |
| Nisar et al. (2020) | Bernal Framework for adaptation | Translation, cognitive interviewing, field testing, questionnaires |
| Suchman et al. (2020) | | Stakeholder engagement, workshops, training, participatory research |
| WHO (2022) | | Situation analysis, stakeholder engagement |
| Zafar et al. (2014) | | Focus groups, interviews, observations, training |

moving from efficacy to effectiveness in the real world) and programme drift (the intervention deviates from its manualised or implementation protocols) (Atif et al., 2022). Different models and methods for training were discussed, including a train-the-trainer or cascade model and technological or digital solutions.

Atif et al. (2017) developed a cascade model to help support the sustainability of the THPP in Pakistan and India. In this model, a specialist 'master trainer' (mental health specialist) trained and supervised a group of non-specialist trainers (university graduates), who in turn provided training and supervision to groups of peers. The cascade model supports sustainability by requiring a minimal number of specialist workers. The model worked differently across the two contexts to ensure intervention quality and sustained motivation of the peer workers, with frequent field supervisions in Pakistan and supervisions to discuss audio recordings of previous sessions in India (Atif et al., 2017). Atif et al. (2019a) evaluated how this model was working 5 years after it was initially implemented. They found that factors contributing to sustained motivation of peer volunteers included altruistic aspirations, enhanced social standing in the community, personal benefits to themselves and possibilities for other avenues of employment. Challenges included demotivation due to uncertainty about the programme's future, increased requirement for financial incentivisation and logistics of organising groups in the community. To further aid the training and support of the cascade model, Atif et al. (2022)

developed a mobile application to support peers who were delivering the THP. The app contained an integrated training module, which reduced the burden on specialist and non-specialist staff to provide training.

Davies et al. (2022) highlight how ongoing, high-quality supervision is essential for the sustainability of community-led programmes. By utilising structured frameworks like the EQUIP initiative, supervision transitions from an administrative check-in to a continuous learning process. This helps prevent skill decay and adapts interventions to remain both effective and culturally relevant.

### Stakeholder engagement

Thirteen information sources included the 'stakeholder engagement' sustainability strategy (Perinatal Mental Health Project, n.d.; Mental Health Innovation Network, n.d.b; UNFPA/WHO, 2009; The Partnership for Maternal, Newborn, and Child Health, 2014; PRIME, 2016; Davies et al., 2019; Honikman and Field, 2020; McNab et al., 2021; Davies et al., 2022; Sarkar et al., 2022; Manolova et al., 2023; Obonyo et al., 2023; Owais et al., 2023). This theme covered engagement with policymakers, people with lived experience, healthcare workers and community groups. For example, Davies et al. (2022) described that a deep participatory approach involving counsellors in the early stages of intervention development is considered important for fostering a sense of ownership and empowerment among them. This process can, in turn, promote

greater acceptance and understanding of the intervention within the community and enhance its long-term sustainability. Additionally, Obonyo et al. (2023) discussed their perspectives on the mental health of pregnant and parenting adolescent girls in Kenya with policymakers, which led to a proposal for comprehensive, adolescent-centred policies and programmes.

The successful sustainability of perinatal mental health interventions in LMICs is dependent on health system factors, particularly workforce capacity and functional referral pathways, which need sufficient stakeholder engagement. Despite the high burden of disease, mental health is often deprioritised on national health agendas, with resources frequently diverted to communicable diseases and visible physical health outcomes (The Partnership for Maternal, Newborn, and Child Health, 2014). Sarkar et al. (2022) noted that in Uganda, the public health system prioritises physical health outcomes over invisible perinatal mental health issues, resulting in insufficient financial and human capital for integration. The sustainability of task-sharing interventions is threatened by the excessive workloads of frontline workers. Atif et al. (2017) highlighted that in Pakistan, LHWs are frequently drafted into priority programmes for polio and dengue, leaving limited capacity for mental health care. Furthermore, effective integration requires robust and reliable mechanisms for escalating care. Owais et al. (2023) found that in private clinics in urban Pakistan, fragmented links to public tertiary facilities prevented the effective referral and management of mothers with severe depression. Sufficient stakeholder engagement may help mitigate these challenges.

## Discussion

### Overview of the research and grey literature

Overall, a total of 42 information sources were identified, mostly published between 2020 and 2023, with 2023 being the most common year. Most sources focused on countries in Asia, particularly Pakistan. The Thinking Healthy Programme (THP) was the most frequently discussed intervention. Terminology used across the literature is inconsistent. Key concepts such as 'scale' are seldom defined, and related terms like 'spread' are absent (Yamey, 2011). These gaps reflect deeper disciplinary divides and call for more interdisciplinary research that incorporates perspectives and theoretical frameworks from implementation science, health economics and organisational studies (e.g., Rogers, 2003). Future research could consider greater interdisciplinary collaboration with implementation scientists to develop specific theoretical frameworks for implementing perinatal health interventions, acknowledging contextual challenges and maximising impact in LMIC settings (Kemp et al., 2019).

It is also likely that significant insights are being generated outside the academic domain. Much of the work on scale is led by non-governmental organisations and development agencies, with findings often documented in grey literature or unpublished internal reports (Spicer et al., 2020). As such, the peer-reviewed literature may offer only a partial view of current strategies and challenges.

### Strategies used and identified gaps

The results of this scoping review highlight that dominant themes in the literature, task-sharing (20 information sources) and integration (20 information sources), are closely aligned with current global policy recommendations in perinatal mental health (WHO, 2022).

While this alignment is expected, how these strategies are implemented varies across settings, underscoring the importance of local context and resource allocation in implementing and scaling interventions. For example, whether the intervention was delivered by non-specialist providers, CHWs or peers depended on the capacity of the local healthcare services and the type of intervention. Additionally, task-sharing interventions may be less effective in certain settings, reflecting challenges around misinterpretation of training and behavioural characteristics of individuals providing the interventions (Davies et al., 2022). Therefore, contextually tailoring implementation strategies may be more successful if factors such as local cultural norms, accessibility and personnel characteristics are carefully considered (Munodawafa et al., 2018).

Adaptation was also frequently addressed (17 information sources). While these studies provide rigorous detail on the adaptation process itself, they often lack consideration for how those modifications impact the intervention's potential to be scaled, spread and sustained across diverse settings. However, there has been a marked increase in publications on this topic, particularly centred on the THP in Asian contexts (e.g., Atif et al., 2017, 2019b; Boran et al., 2023a, 2023b). Figure 3 provides an example of how this intervention has spread and scaled. This concentration of work around the THP, while impressive, highlights a gap in the literature, suggesting a need for broader investigation into a wider range of interventions, models and contexts.

Sustainability is rarely addressed in depth, potentially due to the practical and methodological difficulties of achieving and measuring it over time (Scheirer and Dearing, 2011). This challenge may contribute to a publication bias where unsuccessful or incomplete initiatives are underreported, limiting opportunities to learn from failure (Thornton and Lee, 2000). Additionally, information regarding the outcomes of the strategies (if they 'worked' and for how long) was not always described in detail. Future mixed-methods and longitudinal work should be undertaken to explore what did and did not work and what we can learn from this.

Future work must also acknowledge that specific subpopulations, such as adolescents, women experiencing intimate partner violence and those in humanitarian settings, face elevated risks for perinatal mental health within LMICs. Therefore, in addition to contextually tailoring implementation strategies to the local cultural and health system environment, it is imperative to prioritise strategies that align with the specific, expressed priorities of these diverse groups.

### Limitations

This scoping review has several limitations. First, only studies published in English were included, which may have led to the exclusion of relevant research published in other languages. Second, although grey literature was included, the search was not exhaustive. As a result, key insights may have been omitted. Third, while usually not included in scoping reviews, a formal quality check was not conducted, and this review mixes peer-reviewed studies with grey literature. Therefore, the findings should be viewed with caution as they may rely on data of varying rigor. Additionally, the scoping review lacks comparative analysis across settings and subpopulations. This could be an interesting avenue for future research in this area. Finally, there were differences in how much detail was described across the information sources. While all included sufficient detail to be included, many information sources did not specifically focus on discussing and evaluating a scale, spread or sustainability strategy but rather discussed these

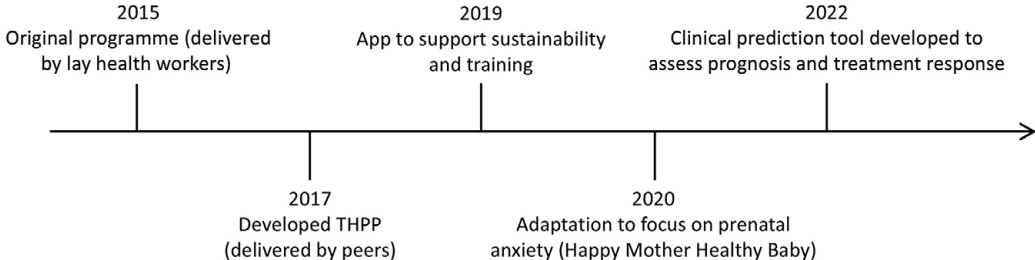

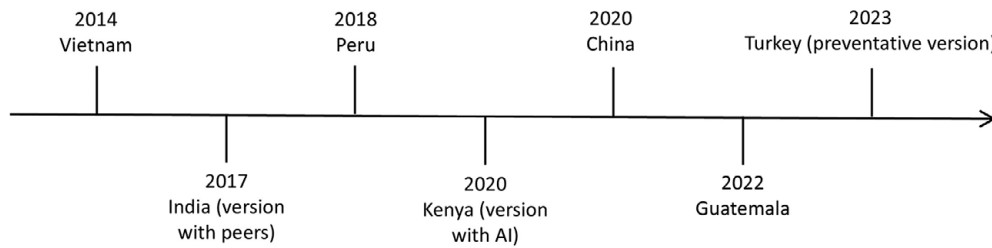

**Figure 3.** Case example: Thinking Healthy Programme.

alongside their other aims and objectives. Comparative analysis across programmes could illuminate strategies that work across diverse settings.

## Conclusion

Taken together, efforts to scale, spread and sustain perinatal mental health interventions in LMICs rely on several interlinked strategies. Diversifying the workforce through task-sharing with non-specialists addresses staff shortages and improves cultural alignment, but requires strong, ongoing supervision to prevent losses in intervention quality and to avoid overburdening low-paid workers. Integrating interventions into routine care, such as maternal and child health services, can increase access to mental health support, though this is often hindered by fragmented systems and competing clinical priorities. Successful integration requires holistic health system strengthening rather than isolated programme add-ons. Cultural and contextual adaptation also plays a central role in spreading interventions across settings, but needs to balance fidelity with locally meaningful delivery. Participatory co-design ensures adaptations enhance, rather than dilute, impact. Digital tools, such as mental health apps and training platforms, offer new avenues for scalable training and supervision, though challenges such as digital exclusion and reduced human connection remain. Sustaining interventions over time depends on early, genuine engagement with communities and policymakers, ensuring alignment with national priorities and long-term funding structures. However, this type of engagement takes time.

Overall, scale, spread and sustainability function as interconnected processes that work together to embed innovations within the health and care systems (Scarbrough and Kyratsis, 2022). Moving beyond isolated projects will require shared understanding of terminology, interdisciplinary collaboration and a shift from proving 'what works' to understanding 'how it works at scale' within complex health systems. Addressing these strengths, challenges and implications is pivotal in translating innovative complex interventions into equitable, sustainable and scalable improvements in perinatal mental health outcomes across LMICs.

**Open peer review.** To view the open peer review materials for this article, please visit http://doi.org/10.1017/gmh.2026.10198.

**Supplementary material.** The supplementary material for this article can be found at http://doi.org/10.1017/gmh.2026.10198.

**Data availability statement.** Data sharing is not applicable – no new data are generated.

**Author contribution.** K.R.M.S. and S.H. conceptualised the project. K.R.M.S. led the data curation, investigation, methodology, funding acquisition and supervision. M.K., M.M. and G.P.K. were involved in data curation and investigation. M.K. and M.M. were also involved in project administration. M.M., S.M. and S.H. provided input regarding the methodology. L.A. supported the data curation and project administration. K.R.M.S., M.K. and M.M. led on the writing of the original draft, which was then reviewed and edited by all authors. All authors agreed to submit the manuscript, read and approved the final draft and take full responsibility for its content. All authors had access to the data.

**Financial support.** This project was funded by Internal Funds from City, University of London, through the Policy Support Fund (Project ID: 48216BT).

**Competing interests.** The authors declare none.

**Declaration of generative AI and AI-assisted technologies in the writing process.** During the preparation of this work, the authors used Microsoft Copilot in order to help reduce the number of words in the abstract and the two final concluding paragraphs. The prompt used included the posting of the original full paragraph written by the first author. Then Copilot was asked to 'reduce the word count to 200 words and make the writing more concise'. After using this tool/service, the authors further reviewed and significantly edited the content as needed and take full responsibility for the content of the publication.

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
