## [Reviewer Report]

Thank you for the opportunity to review this paper. It was an engaging read. The manuscript presents a timely and much-needed scoping review on scale, spread, and sustainability strategies for perinatal depression, which fills a critical gap in existing literature. The authors have done a commendable job in developing a well-written and informative piece. However, I have a few comments and suggestions that may help strengthen the paper further.

Introduction

• The introduction is well written but appears somewhat broad, beginning with general discussions on mental health before narrowing to perinatal depression. While this approach is not problematic, the paper would benefit from expanding the third paragraph, which discusses existing gaps in perinatal mental health care. The authors note that there is evidence on interventions but provide only a general statement. It would strengthen the introduction to more specifically outline what evidence currently exists on perinatal depression interventions and to elaborate on the specific gaps in the literature.

• Lines 25–27: Would it be possible to rephrase this sentence? The term “non-study” environment is somewhat unclear. Additionally, the authors refer to a “problematic translation of intervention into scalable and sustainable programmes.” It would be helpful to elaborate briefly on what these problems or challenges entail. Are they due to the short-term nature of projects, limited advocacy, lack of political buy-in, cost-related constraints, or other factors? Or has this issue not been explored at all?

Methods

• In the Methods section, the authors mention searching websites and grey literature. Could more details be provided on how the websites were systematically searched? Additionally, please clarify what types of materials were included as grey literature.

Results

• The section on “characteristics of the included information sources” and “scale, spread, and sustainability strategies” is somewhat difficult to follow because of the long list of references. Expanding Table 2 to include additional details such as study sources and other key information, would allow readers to refer to the table for specifics, while the text could focus more on summarizing key findings for improved readability.

• “Diversity of Workforce”: The distinction between task sharing and task shifting is very well articulated. It would be useful to indicate how many studies reported each of these approaches to help readers understand which is more commonly applied or recommended. Have these studies also described when and under what circumstances each approach is most appropriate? If such information is not available in the results, discussing this in the Discussion section would add valuable insight.

• “Information sources, especially those written more recently, discussed the importance of using methods from implementation science” – This point is somewhat unclear. Could the authors specify which methods from implementation science are being referred to and explain their relevance?

• In the paragraph discussing the example from Uganda, it would be helpful to elaborate briefly on what this framework entails—its purpose and how it connects to the scaling strategy. Is this framework primarily a tool for mapping challenges and solutions to scaling up, or can it itself be considered a strategy?

• While this may seem to contrast with my earlier suggestion regarding referencing, I would appreciate being able to trace studies that have applied different adaptation methods. Could the relevant references be listed here or included in a table?

• “For example, Suchman et al. (2020) adapted a parenting intervention developed in the United States for use in South Africa.” It is unclear what the authors intend to highlight with this example. Are they emphasizing that the spread strategy involved adapting an intervention to address perinatal depression specifically, or simply illustrating adaptation to a new context? If it is the latter, clarification would help, especially since a parenting intervention may not directly address perinatal depression. A brief explanation would help prevent confusion.

• In the section “Training, supervision, and support,” the discussion appears heavily focused on training. A more balanced analysis that includes adequate detail on supervision and support would strengthen this section.

Discussion

• “Most sources focused on countries in Asia, particularly South Asia.” Based on the table, most studies seem to be from Pakistan. It may be clearer and more accurate to specify the country rather than referring to the region more generally.

• “Adaptation was also frequently addressed (17 information sources). However, this was typically in isolation and with limited exploration of how interventions are modified and scaled across diverse settings and contexts.” Since intervention adaptation studies often describe detailed processes, it would be helpful to clarify what is meant by “adaptation was typically done in isolation.” Does this refer to the lack of integrated frameworks or the absence of consideration for scaling during adaptation?

• Lastly, while the discussion effectively identifies existing gaps, it would be valuable to include broader reflections on the different strategies for scale, spread, and sustainability, highlighting their respective advantages, challenges, and implications. This would provide a more integrative synthesis and strengthen the overall contribution of the review.

---

## [Reviewer Report]

I think adding the category of lay counselor would enhance the quality of the article, but would not be essential to the overall article.

Overall, this review provides a thorough background of the urgent need for adequate perinatal mental health intervention in low and middle -income countries (LMICs) that are scalable, easily adopted and spread, and sustainable and thus, makes a significant and important contribution to the field of Global Mental Health and is deemed by this reviewer as suitable for this journal.. 

The introduction succinctly describes the current state of need for strategies that effectively and efficiently address this need and emphasizes that these strategies need to be scalable, adoptable (spreadable) and sustainable. They cite numerous references in an attempt to define their case for defining these strategies.

The Methodology section was well constructed and organized. The authors professionally describe their methodology in a way that could be replicable. 

One suggestion for a non-essential improvement offered by this reviewer falls under the Methodology’s scale strategies- diversified workforce sub-section. I think there is an important omission in the classifications of workforces examined; and that is the classification of lay counselors. Task sharing and task-shifting strategies often produce great results in research but often are found to be unsustainable due to professionals already being overburdened by other professional medical responsibilities. CHW’s are already overburdened and in short supply. Lay counselors are in a separate category than peers as lay counselors may not necessarily have shared life experiences. Lay counselors are, however, professionally trained lay people, drawn from the communities they live and work in, and trained in specific mental health interventions. While they have their own problems with sustainability, due to lack of ongoing funding, they can play an important role in providing community mental health interventions. The use of lay counselors is well described in the current literature. 

The Discussion section brings up important current shortcomings to sustainability as mentioned in my comments about lay-counselors (above)

Overall, this is a clear review of current efforts toward providing mental health interventions in an underserved population and the review provides clarity for moving forward using agreed upon definitions .

---

## [Reviewer Report]

Strategies for spreading, scaling, and sustaining perinatal mental health interventions in Low- and Middle-Income Countries (LMICs) – A scoping review and thematic synthesis

Thank you very much for the opportunity to review this manuscript. It is very well written and the methodology is sound. I have provided comments on the manuscript below.

I believe that this scoping review is important given the dearth of studies on PMH. I would, though, like the purpose of the manuscript to be clearer including what the reader should take away from it. What specific gap is this review aiming to fill, how is the information useful, and how can it be applied? The writing is excellent, but I believe that the manuscript will be more useful if the authors consider their audience and think about how they would like them to use the information and consider conveying the information in a more utilitarian manner.

Impact Statement. Row 10: Do you want to include the term ‘evidence-based’. Also adopted by the WHO? There is a foundation for this that has been implemented. Impact statement doesn’t reflect that.

Introduction:

First paragraph: Interesting to focus on mental health conditions in general – the mention of strain on health systems due to balance of communicable and non-communicable seems insufficient in this current context where, at baseline there was already limited funding available for mental health – now exacerbated by foreign aid cuts. In addition, persistent multi-level stigma inhibits policies, systems and practice that support the mental health rights of individuals. In regards to PMH, the intersection of gender and mental health (and other factors for high-risk populations like HIV, GBV, etc.) also creates challenges to influence investments and attention in this area.

Second paragraph: Suggest providing a comparison for range in low-income countries in comparison to high-income countries for further context. The language is also limited to just the impact on the mother and infant in terms of the fetal development and the quality of their relationship. As the authors are aware, the harmful impacts extend well beyond this including into the adolescent period, with potential intergenerational impacts, while also influencing families. Suggest expanding this language here. This language will help readers to understand the justification for why they should care about scale-up.

Row 15: Please add a semi colon at the end of point 2.

Final paragraph in the introduction: Suggest adding information on who should use this information and how they should use it.

Generally in introduction: The WHO’s Thinking Healthy program is well established and I am wondering why the authors did not at minimum mention this in terms of its existence, and its use globally? Suggest including a baseline description at least of the resource itself, noting that PMH interventions implemented by Member States have been a priority within mental health efforts at WHO.

Methods:

Data appraisal. Given grey literature was also included within the scoping review, if a data appraisal was not conducted, I suggest adding information in the discussion on the grey literature in particular and the weight with which the findings should be taken without a data appraisal and if and where there were any specific concerns in this regard.

Results:

As a reader, the information is not very accessible in its current form by simply listing the authors of papers under each of the categories for the first few pages. I understand the need to do this, but I am wondering if there is a way to do it that makes the actual information within the manuscript more accessible to the reader.

The lead author of this scoping review has led authorship of an excellent manuscript examining results from a very interesting CHIME study in Gambia regarding a singing intervention. Why is that not included within the results? It would fit within the tool and method development nicely and deserves a brief summary within this section. Did the methodology perhaps miss some studies?

Under sustainability, the low priority of PMH within the already weak systems seems like it should be a factor that should be considered within the results. At the very least, even without any of the manuscripts specifically addressing systems factors, I suggest authors note this gap in the results through noting that none of the manuscripts mentioned national funding, human resources for PMH, data systems, etc.

Discussion:

Strategies and identified gaps. It is also important that the authors note that specific subpopulations of girls and women experience elevated risk within LMIC. In addition to mentioning contextual tailoring implementation strategies, it should also be noted that strategies that align with the specific priorities of these diverse subpopulations should be a priority.

Limitations:

The scoping review also lacks comparative analysis across settings and subpopulations.

Table 1;

Population: Noting that this states women. Does this mean that studies with girls under 18 years of age were not included, or should the population be expanded to include ‘girls and women’?

Intervention: Where is stillbirth in this?

---

## [Editor Report]

Dear Authors,

I am happy to share with you that the independent review of your manuscript is over, and the reviewers have provided detailed comments. I encourage you to kindly go through them and address as you see relevant, and resubmit the manuscript for further processing.

Looking forward to receiving the revised manuscript,

With best wishes,

Thomas

---

## [Editor Report]

Dear Authors,

Thank you for revising the manuscript and addressing the reviewers' comments satisfactorily.

I am happy to share with you that your article has been accepted. Thanks for choosing the ‘Cambridge Prisms: Global Mental Health’ for showcasing your work. I hope you will consider this journal for publishing your future work.

Best wishes,

Thomas